# The Association between Breakfast Macronutrient Composition and Body Weight, in Preadolescence: An Epidemiological Study among Schoolchildren

**DOI:** 10.3390/children9121960

**Published:** 2022-12-13

**Authors:** Stamatia Kokkou, Venetia Notara, Aikaterini Kanellopoulou, George Antonogeorgos, Andrea Paola Rojas-Gil, Ekaterina N. Kornilaki, Areti Lagiou, Demosthenes Panagiotakos

**Affiliations:** 1Department of Public and Community Health, Laboratory of Hygiene and Epidemiology, School of Public Health, University of West Attica, Alexandras Avenue 196, 115 21 Athens, Greece; 2Department of Nutrition and Dietetics, School of Health Sciences and Education, Harokopio University, Thiseos 70, 176 76 Athens, Greece; 3Department of Nursing, Faculty of Health Sciences, University of Peloponnese, Karaiskaki 70, 221 00 Tripoli, Greece; 4Department of Preschool Education, School of Education, University of Crete, 741 00 Gallos, Greece

**Keywords:** children, overweight/obesity, breakfast, protein, lipids, macronutrients

## Abstract

Over the last few decades, childhood overweight and obesity tend to reach epidemic proportions. The present study aimed to examine whether the composition of breakfast, through the prism of its macronutrient content, could play a part in the development of excess body weight in children. A sample of 1728 Greek students from 47 primary schools was enrolled for the purposes of this study. Their weight and height were measured and their Body Mass Index was calculated and classified according to the International Obesity Task Force criteria. Their dietary habits, breakfast foods of choice, and physical activity were assessed through the use of a self-completed questionnaire. Further assessment of breakfast composition was carried out in order to evaluate its macronutrient content. Initial analysis, which was only energy-adjusted, showed a negative association between body weight and breakfast protein content and a positive association for lipid content. After further adjustment for age, sex, level of adherence to the Mediterranean diet, and physical activity, there was a significant negative association between breakfast protein and excess body weight in children (*p* = 0.029), as well as a significant positive association of breakfast lipid content (*p* = 0.028). Breakfast macronutrient content seems to have an effect on body composition in children, independently of overall adherence to the Mediterranean diet and physical activity, however further research is needed in order to elucidate potential pathways.

## 1. Introduction

According to the World Health Organization (W.H.O.), a variety of chronic health conditions that appear during adult life, such as cardiovascular disease (CVD), type II diabetes mellitus, musculoskeletal disorders, and some types of malignancies (i.e., colon, breast, endometrial), seem to be related to the excessive body weight in childhood [1]. It is already known that within the last five decades the prevalence of childhood and adolescent overweight and obesity has almost increased fivefold in both sexes [2]. Specifically, in 2016 almost 1 in 5 children and adolescents aged 5–19 years were overweight and about 1 in 15 were obese, worldwide [2]. Considering this evidence, many organizations have been alerted, highlighting the importance of addressing the issue of excessive body weight early in life, by focusing on lifestyle patterns and dietary habits and promoting strategies to improve those aspects before high adiposity levels become established [3,4,5]. However, the pathways through which prevention can be achieved vary and need to be thoroughly studied. The effect of breakfast meals is one such pathway, which has concerned the scientific community, especially in young children. However, research has been mostly focused on the habit of breakfast consumption versus breakfast skipping, and evidence suggests that there is either protective or no effect of frequent breakfast consumption in the occurrence of overweight and obesity in childhood and adolescence [6,7,8,9,10,11]. Moreover, it has been hypothesized that breakfast dietary patterns may play a key role in body adiposity levels and markers of cardiometabolic health, indicating it is the meal’s composition, rather than its consumption, that may affect the outcome [10,12,13,14]. Nevertheless, little evidence exists suggesting a potential effect of breakfast composition under the prism of its macronutrient content and whether this approach might elucidate its overall effect on body composition [15], thus allowing us to better understand and prevent overweight and obesity early on in life. 

Under the aforementioned considerations, the aim of the present study was to investigate the potential association between weight status, and breakfast composition, in terms of its macronutrient component, in Greek children aged 10–12 years. 

## 2. Materials and Methods

### 2.1. Design

Regarding study design, this is a cross-sectional, population-based, observational one. It was carried out in the greater metropolitan Athens area, in Heraklion, the capital city of the Island of Crete, and in three main counties of the Peloponnese peninsula. The participants were recruited and measured during the academic years 2014–2015 and 2015–2016. These districts were chosen in particular, because they represent large urban and rural municipalities; thus, a representative sample was obtained. The Greek Ministry of Education provided a list of schools and the selection process was held using random sampling, leading to a total of 47 schools. Specifically, 32 out of the 47 schools were in the Attica region, the capital city of Greece, whereas 5 were in the capital city of Crete Island, Heraklion, while the rest 10 were in three large cities of the Peloponnese peninsula, (i.e., Sparta, Kalamata and Pyrgos).

### 2.2. Participants

The study sample consisted of 1728 schoolchildren (933 girls and 795 boys), aged 10–12 years. For the purposes of the present analysis, only students with available nutrient information regarding breakfast consumption were studied; thus, the working sample consisted of 1521 schoolchildren from 47 schools from 5 regions of Greece. No significant differences regarding age, sex, and region of residence between the children included and those excluded (n = 207) in dietary analyses (*p*’s > 0.10). Among schools, the participation rate ranged from 95% to 100%.

### 2.3. Power Analysis

The current sample was sufficient to evaluate the effect size measures’ differences between the study groups of 15% on the prevalence of overweight/obesity, achieving 80% statistical power at <5% level of significance.

### 2.4. Measurements

A questionnaire was given to the children, which they completed by themselves, with trained field researchers and educators present, to clarify any misconceptions. The study’s questionnaire, among others, included: information on children’s socio-demographic characteristics (age, gender, parental financial status, and educational level,), a validated Food Frequency Questionnaire (FFQ) [16], as well as a validated questionnaire on physical activity [17]. The frequency and type of breakfast were assessed in detail. Specifically, the frequency was estimated with the use of a scale of five possible answers, ranging from 1 = never/almost never to 5 = every day. Their usual breakfast choices were also assessed through the questionnaire, where they could mark any number of answers, i.e., milk, yogurt, cereals, fruit juice, honey/jam, bread/rusk, butter/margarine, cake/tsoureki/koulouria (a type of Greek cookies) and chocolate milk. Breakfast composition in terms of macronutrients was evaluated through the use of the “Composition tables for Greek foods and recipes” [18]. Protein, carbohydrate, and lipid content were calculated for each answer’s serving size, using mean content for each macronutrient for answers containing more than one item. Macronutrient content was measured in grams. The macronutrients’ intake estimation was furtherly energy-adjusted. For the quality estimation of the sample’s dietary habits in total, the KIDMED Score was used, a scale ranging from −4 to 12 [19]. Up to date, the KIDMED Score is the most widely used for the assessment of adherence to the Mediterranean diet, which is the recommended dietary pattern for health promotion for the Greek population, among children and adolescents. Children’s data concerning exercise specifics and physical condition were determined by their engagement in extracurricular activities, such as partaking in athletic teams, playing with peers, running, swimming, etc., on either a daily basis or weekly, and was classified into a binary (no or yes) variable.

### 2.5. Body Status Assessment

Children’s anthropometric characteristics, namely height in m and weight in Kg, were determined using standard procedures, and thus Body Mass Index (BMI) was computed as body weight in Kg divided by height in m squared. Body weight status was classified using the International Obesity Task Force (IOTF) cut-off criteria for Body Mass Index (BMI) for the specific age group and appropriately for each gender, which links the Index at 18 years (16, 17, 18.5, 25, and 30 kg/m^2^) in child centiles [20]. For the data analyses, children living with overweight and the ones living with obesity were combined due to the small sample size within each of those categories.

### 2.6. Statistical Analysis

Continuous variables are displayed as mean value ± standard deviation, while categorical variables are displayed as both absolute frequencies (n) and relative (%) ones. Kolmogorov-Smirnov test was used to examine if the quantitative variables concerning protein, carbohydrate, and lipid content, of breakfast food choices, were normally distributed. After normal distribution was confirmed, Student’s *t*-test was used to compare means between the two weight status groups for each macronutrient. For the assessment of the association of breakfast composition, in terms of macronutrients, on overweight and obesity status, logistic regression was performed for protein, carbohydrate, and lipid contents, separately. The results are displayed as odds ratios (OR) along with their corresponding 95% confidence intervals (95% CI). Potential confounders, i.e., measurements that have shown a significant association with weight status according to the literature (i.e., age, sex, KIDMED Score, and physical activity) were included in the regression models to further evaluate the independent role of macronutrients studied in relation to the outcome. Parental body weight was not associated with the exposure factors (i.e., macronutrients), and therefore, was not considered as a confounder. For the evaluation of multicollinearity among the independent variables, variance inflation factor and tolerance were used. All analyses were conducted using Stata 14.0 (M. Psarros and Assoc., Sparti, Greece) at the 5% significance level.

### 2.7. Bioethics

The Institute of Educational Policy, of the Ministry of Education and Religious Affairs, approved the study (code of approval F15/396/72005/C1), which was conducted in accordance with the Declaration of Helsinki (1989). The staff of the institutions (teachers and administration), as well as participants and their parents, were thoroughly informed about the aims and procedures of the study. Before the fulfillment of the questionnaires, parents signed informed consent.

## 3. Results

Children’s characteristics, along with the distribution of breakfast choices, in total and by weight status category are displayed in Table 1. The mean age of the sample was 11.21 ± 0.783 years and 54.6% were girls. Overweight and obesity had a higher prevalence in boys, compared to girls (*p* = 0.006). Regarding breakfast choices, 78.2% of the children regularly consumed white milk, 10% yoghurt, 67.7% cereals, 27% fruit juice, 30.6% either honey or jam, 29.1% bread or rusk, 11% butter or margarine, 21.8% either cake, tsoureki or koulouria and 15.1% chocolate milk. The only significant association between breakfast choices and children’s weight status shown was that of white milk consumption, which was more common for children of normal weight (*p* = 0.018). No significant association was observed between weight category and breakfast either protein, carbohydrate, or lipid content (*p* > 0.005). Finally, the mean KIDMED Score was significantly higher in the normal weight group (*p* = 0.010), whereas children living with overweight/obesity were less likely to engage in any type of physical activity (*p* = 0.019).

Preliminary, energy-adjusted analysis showed a statistically significant negative association between the occurrence of overweight/obesity in preadolescence and the quantity of breakfast protein content (*p* = 0.009), along with a statistically significant positive association with the quantity of breakfast lipid content (*p* = 0.005). A marginally non-significant association was observed for breakfast carbohydrates (*p* = 0.054) (Table 2).

In Table 3 the results from the multiple logistic regression models, evaluating the association of breakfast macronutrients on children’s weight status, are presented. In model 1, which was only energy-adjusted, a significant inverse association was revealed between protein content and children’s overweight/obesity status (OR: 0.962, 95% CI: 0.933–0.992), as well as a positive association with lipid content (OR: 1.028, 95% CI: 1.009–1.048); however, no association was observed concerning carbohydrate content and body weight status (OR: 0.999, 95% CI: 0.993–1.004). In model 2, when age and sex were taken into account, the aforementioned associations remained similar. In the final model (model 3), after adjustment for age, sex, physical activity status and KIDMED score, protein and lipid content of children’s breakfast choices were still significantly associated with their body weight status. Additionally, excess body weight was statistically associated with children’s gender, as boys were more likely there was a negative significant association between physical activity and weight status category with children’s weight status (OR: 0.727, 95% CI: 0.553–0.955).

## 4. Discussion

The present work aimed to examine the association between the composition of breakfast, in terms of its macronutrient content, and the occurrence of overweight/obesity in Greek children of 10–12 years of age. After accounting for potential confounders, a significant association was shown between higher protein content and with lower probability of living with either overweight or obesity. Additionally, the amount of lipids included in breakfast foods was associated with increased odds of suffering from overweight/obesity. However, breakfast carbohydrates did not appear to have any association with weight status in children.

The effect of different macronutrients on body weight, independently of the caloric load, is an issue dividing researchers for the past decade. Especially, from a metabolic viewpoint, evidence suggests that diets of different macronutrient compositions affect adiposity dissimilarly, as their impact on appetite, thermogenesis, and metabolism vary and it also depends on a person’s microbiota and genotype [21]. Existing literature provides distinct suggestions on which macronutrient research should focus, as there are indications that an increase in carbohydrates yields an increase in body weight, whereas the same percent elevation of diet lipids leads to lower obesity prevalence [22]. However, there is also evidence that only dietary fats, specifically polyunsaturated fatty acids, have a significant inverse association with childhood obesity [23]. Finally, there are scientific leads pointing to the interactions between genes and diet, mostly concerning energy expenditure and lipid metabolism, and subsequently weight management and adipose tissue accumulation [24].

A potential mechanism explaining our results could be that higher breakfast protein content might be able to assist maintain one’s normal body weight, through increasing satiety levels after consumption. There are a few previous studies and clinical trials that have examined the role of breakfast protein on weight in childhood and adolescence. One clinical trial, conducted in 2013, on adolescent girls living with overweight/obesity compared the effect of a high-protein breakfast, with a normal-protein one and breakfast skipping, and concluded that the first improved satiety and overall diet quality [25]. Furthermore, a Chinese study evaluated the effect of breakfast protein on weight management in adolescents living with obesity, and showed that it promoted weight loss, possibly by regulating satiety [26]. The satiety path has been proposed by various studies, mostly researching adult participants, as a possible way through which protein acts beneficially on body weight regulation [27,28,29].

Furthermore, adherence to the Mediterranean diet, evaluated through the KIDMED Score, as well as physical activity, have also been previously studied, concerning their impact on the body composition of children and adolescents. A Spanish study of 189 children with mean age of 10.98 ± 2.89 years examined the effect of both aforementioned factors, along with sedentary lifestyle and sleep, and reported no significant differences in children’s body weight status [30]. Another study from Italy researched a sample of 690 children at 9–11 years of age and found no significant association between their weight status and either physical activity or KIDMED Score [31]. Those results are partially in accordance with ours, however, a previous study from Greece, using a sample of 9- to 13-year-old children, showed that higher adherence to the Mediterranean diet was inversely associated with excess body weight, although they pointed out that physical activity seemed to have a more significant effect [32].

There are a few limitations that should be taken into account, when evaluating the results of the present study. Mainly due to the observational study type, no temporal relationship and, hence, causal inferences can be made. The sample was derived only from urban areas of Greece; however, the representativeness of the study could be considered high given that the participating areas have similar characteristics with all other Greek areas, the study sample was large and a stratified random sampling scheme at the school level was used. Another limitation emanates from the fact that some of the breakfast foods of the present study are common in Greece, while eating habits and breakfast of choice in other countries, and thus overall breakfast composition, may significantly differentiate. Finally, to reduce reporting bias in the children’s questionnaire, trained investigators were present throughout the whole procedure of completing the questionnaire in schools to address any potential misconceptions, increasing the validity of the given responses.

## 5. Conclusions

The present work showed a statistically significant association between overweight and obesity in preadolescence in relation to breakfast composition in terms of macronutrients. Children of normal weight consumed higher amounts of protein and lower amounts of lipids in their breakfast meal, compared to ones living with either overweight or obesity, after adjustment for several confounders. Physical activity was also significantly related to children’s weight status, when evaluated along with the breakfast macronutrient content, although adherence to the Mediterranean diet was not. This evidence points towards a great need for health promotion interventions focusing on the necessity and quality of breakfast for children and adolescents, in order to foster a healthy lifestyle, however further research is required.

## Figures and Tables

**Table 1 children-09-01960-t001:** Characteristics of the children by body weight status.

Characteristics	Overall (n = 1521)	Normal Weight (n = 1100)	Overweight/Obese (n = 421)	*p*
Age (years)	11.21 ± 0.783	11.23 ± 0.794	11.16 ± 0.752	0.083
Gender				**0.006 ***
Boys	690 (45.4%)	475 (43.2%)	215 (51.1%)	
Girls	831 (54.6%)	625 (56.8%)	206 (48.9%)	
Breakfast choices (yes)				
White milk	1189 (78.2%)	877 (79.7%)	312 (74.1%)	**0.018 ***
Yoghurt	152 (10%)	115 (10.5%)	37 (8.8%)	0.332
Cereals	1030 (67.7%)	753 (68.5%)	277 (65.8%)	0.321
Fruit juice	411 (27.0%)	304 (27.6%)	107 (25.4%)	0.383
Honey/Jam	466 (30.6%)	347 (31.5%)	119 (28.3%)	0.214
Bread/Rusk	443 (29.1%)	334 (30.4%)	109 (25.9%)	0.086
Butter/Margarine	167 (11.0%)	113 (10.3%)	54 (12.8%)	0.154
Cake/Tsoureki/Koulouria	331 (21.8%)	228 (20.7%)	103 (24.5%)	0.114
Chocolate milk	229 (15.1%)	165 (15.0%)	64 (15.2%)	0.922
Protein content (g)	12.63 ± 6.25	12.82 ± 6.40	12.13 ± 5.80	0.052
Carbohydrate content (g)	60.26 ± 35.09	60.91 ± 35.60	58.55 ± 33.71	0.240
Lipid content (g)	11.25 ± 8.72	11.13 ± 8.54	11.54 ± 9.20	0.428
Breakfast energy content (kcal)	392.75 ± 224.02	395.11 ± 225.69	386.57 ± 219.75	0.506
KIDMED Score (−4 to 12)	4.64 ± 2.28	4.74 ± 2.28	4.40 ± 2.27	**0.010 ***
Physical activity				**0.019 ***
Yes	1202 (79.0%)	886 (80.5%)	316 (75.1%)	
No	319 (21.0%)	214 (19.5%)	105 (24.9%)	

Data are presented as mean ± standard deviation for quantitative variables and counts (percentages) for categorical. * Level of significance is set at *p* < 0.05. *p*-values marked in bold indicate a statistically significant association.

**Table 2 children-09-01960-t002:** Energy-adjusted logistic regression models for the evaluation of protein, carbohydrate, and lipid content on children’s weight status.

	Odds Ratio	95% Confidence Interval	*p*
Protein content (in g)	0.956	(0.924; 0.989)	**0.009 ***
Carbohydrate content (in g)	0.989	(0.977; 1.000)	0.054
Lipid Content (in g)	1.038	(1.011; 1.065)	**0.005 ***

Macronutrients were entered separately in each model. * Level of significance is set at *p* < 0.05. Energy adjustments were made in all models. *p*-values marked in bold indicate a statistically significant association.

**Table 3 children-09-01960-t003:** Multiple logistic regression models for the evaluation of breakfast’s macronutrient content on children’s weight status, after adjustment for confounders.

	Model 1	*p*	Model 2	*p*	Model 3	*p*
Protein content (g)	0.962	**0.014 ***	0.962	**0.015 ***	0.966	**0.029 ***
	(0.933; 0.992)		(0.933; 0.992)		(0.936; 0.997)	
Carbohydrate content (g)	0.999	0.603	0.998	0.601	1.000	0.870
	(0.993; 1.004)		(0.993; 1.004)		(0.994; 1.005)	
Lipid Content (g)	1.028	**0.003 ***	1.028	**0.004 ***	1.022	**0.028 ***
	(1.009; 1.048)		(1.009; 1.048)		(1.002; 1.042)	
Age (years)	-		0.879	0.080	0.857	0.039
			(0.760; 1.016)		(0.741; 0.992)	
Sex (girl vs. boy)	-		1.351	**0.009 ***	1.379	**0.006 ***
			(1.077; 1.695)		(1.098; 1.732)	
KIDMED Score (−4 to 12)	-		-		0.955	0.082
					(0.906; 1.006)	
Physical activity (no vs. yes)	-		-		0.727	**0.022 ***
					(0.553; 0.955)	

Data are presented as Odds Ratio (95% Confidence Interval). * Level of significance is set at *p* < 0.05. Energy adjustments were made in all models. *p*-values marked with in bold indicate a statistically significant association.

## Data Availability

Data can be made available upon request.

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
