# Peer review of "The Association between Breakfast Macronutrient Composition and Body Weight, in Preadolescence: An Epidemiological Study among Schoolchildren"

_children, 2022, doi:10.3390/children9121960_

Round 1
Reviewer 1 Report
First of all, I would like to thank for the opportunity to revise the manuscript entitled:
„The association between breakfast composition and body weight, in preadolescence: an epidemiological study among schoolchildren”.
This is a well-written article that identifies an important issue in the field of science about relations between weight status and breakfast composition
The title is correct and explains the aim of the current study well, please do not put a dot in the end of the title.
The abstract is well written, it summarizes the article.
The introduction is well-written, the examined indicators are well-summarized. The information about the expansion of overweight and obese children is alarming end every cause has to be assessed.
Ii is not clear in which periods were the participants measured, in years 2014-15 and 2015-16?
The discussion is well-written and logical, well-supported by several articles.
Tables and figure are well-formatted and informative. Please, check in Table 1 the Boys and Girls row, because there are some inaccurate percentages.
Author Response
Replies to Reviewer 1# comments
This is a well-written article that identifies an important issue in the field of science about relations between weight status and breakfast composition
The title is correct and explains the aim of the current study well, please do not put a dot in the end of the title.
The abstract is well written, it summarizes the article.
The introduction is well written, the examined indicators are well-summarized. The information about the expansion of overweight and obese children is alarming end every cause has to be assessed.
Reply: We would like to thank the Reviewer very much for the time spent on our work, as well as the useful comments made that helped us, we believe, to improve the presentation of our findings.
Ii is not clear in which periods were the participants measured, in years 2014-15 and 2015-16?
Reply: The participants were recruited and measured during the academic years 2014-15 and 2015-16. The sentence was rephrased, in the revised ms, as proposed (pls see pg 2).
The discussion is well-written and logical, well-supported by several articles.
Reply: We would like to thank the Reviewer for the positive comment.
Tables and figure are well-formatted and informative. Please, check in Table 1 the Boys and Girls row, because there are some inaccurate percentages.
Reply: The percentages of the Boys and Girls row are referred to the weight status group, and not to the gender group.
Reviewer 2 Report
Dietary nutrition is a key variable for health status of human, so keeping healthy diet should be followed. However, accurate evidence on it is still not enough. It is interesting study in which authors investigated the association between breakfast composition and body weight in preadolescence of Greece. The manuscript is written well but some technical issues should be addressed further.
1. The title seems a bit confusing. Here composition should refer to nutrient composition in breakfast. So it would be better to change title for clear description of study topic.
2. Author said some children were excluded due to missing information on diet. But they did not present the difference between the participants included and those excluded. A supplementary file should be included to show the difference.
3. It is a little confusing about power analysis. What is figure of 15%? It refers to what? In this study, topic is to investigate association of macronutrients in breakfast with overweight and obesity. So sample size estimation should focus on it.
4. For macronutrients intake estimation, are they energy-adjusted? Authors should clarify this issue in the part of methods.
5. Why did authors adjust for KIDMED Score? Some justification should be provided.
6. How did authors consider covariates for adjustment? What covariates do they included? And why? In present model, just age, sex, KIDMED, PA are adjusted. How about family information?
Author Response
Replies to Reviewer 2# comments
- 1. The title seems a bit confusing. Here composition should refer to nutrient composition in breakfast. So it would be better to change title for clear description of study
Reply: We would like to thank the Reviewer very much for the time spent on our work, as well as the useful comments made that helped us to improve the manuscript. We added the term “macronutrient” in the title, in the revised ms (pls see title).
- Author said some children were excluded due to missing information on diet. But they did not present the difference between the participants included and those excluded. A supplementary file should be included to show the difference.
Reply: We have added in the revised version of the paper the following text “No significant differences regarding age, sex and region of residence between the children included and those excluded (n=207) in dietary analyses (p’s > 0.10).”
- It is a little confusing about power analysis. What is figure of 15%? It refers to what? In this study, topic is to investigate association of macronutrients in breakfast with overweight and obesity. So sample size estimation should focus on it.
Reply: The relevant paragraph was rephrased in the revised manuscript (pls see pg 2).
- For macronutrients intake estimation, are they energy-adjusted? Authors should clarify this issue in the part of methods.
Reply: The certain clarification was added in the “Methods” section of the revised manuscript (pls see pg 3).
- Why did authors adjust for KIDMED Score? Some justification should be provided.
Reply: The KIDMED score is the most widely used for the assessment of adherence to the MD among adolescents. A justification was provided in the revised ms (pls see pg 3).
- How did authors consider covariates for adjustment? What covariates do they included? And why? In present model, just age, sex, KIDMED, PA are adjusted. How about family information?
Reply: Covariates were entered in the model under the principle of potential confounding. Family structure was not entered in the final model because it was not a mediator or a moderator in our analyses.
Round 2
Reviewer 2 Report
Authors have addressed my comments, but some minor concerns should be addressed further for improvement of manuscript.
1. About KIDMED Score, authors do not present clearly why it was adjusted in the model. It is considered as it is regarded as total energy? Here, authors considered it as confounder but they should provide justfication.
2. Intake of nutrients is related closely to total energy. Therefore, energy-adjusted intake of nutrients should be considered or adjusting energy as confounder in the model.
3. It is still unclear about confouder selection. If authors think some covariates are not regarded as confounders, they should provide clear statement on reason in the part of statistics.
Author Response
- 1. About KIDMED Score, authors do not present clearly why it was adjusted in the model. It is considered as it is regarded as total energy? Here, authors considered it as confounder but they should provide justfication.
Reply: We thank the Reviewer for the useful comments made that helped us to improve the manuscript. The KIDMED Score was used as a confounder in order to examine the impact of the breakfast macronutrient composition independently of the overall quality of the participants’ dietary habits, expressed as level of adherence to the Mediterranean diet.
- Intake of nutrients is related closely to total energy. Therefore, energy-adjusted intake of nutrients should be considered or adjusting energy as confounder in the model.
Reply: We performed energy-adjusted analyses according to your comment (pls see pgs 1, 4, 5 of the ms). Due to multicollinearity, no differences are observed in the final regression models.
- It is still unclear about confouder selection. If authors think some covariates are not regarded as confounders, they should provide clear statement on reason in the part of statistics.
Reply: Potential confounders, i.e., measurements that have shown a significant association with weight status according to the literature (i.e., age, sex, KIDMEDSCORE, and physical activity) were included in the regression models to further evaluate the independent role of macronutrients studied in relation to the outcome. Parental body weight was not associated with the exposure factors (i.e., macronutrients), and therefore, was not considered as a confounder. Other measurements, like environmental factors (e.g., living environment, etc), were not measured in our study, and thus could not be included in the models (pls see pg 3 of the ms).